# HLA Class II Polymorphism and Humoral Immunity Induced by the SARS-CoV-2 mRNA-1273 Vaccine

**DOI:** 10.3390/vaccines10030402

**Published:** 2022-03-06

**Authors:** Juan Francisco Gutiérrez-Bautista, Antonio Sampedro, Esther Gómez-Vicente, Javier Rodríguez-Granger, Juan Antonio Reguera, Fernando Cobo, Francisco Ruiz-Cabello, Miguel Ángel López-Nevot

**Affiliations:** 1Servicio de Análisis Clínicos e Inmunología, University Hospital Virgen de las Nieves, 18014 Granada, Spain; fruizc@ugr.es (F.R.-C.); manevot@ugr.es (M.Á.L.-N.); 2Programa de doctorado en Biomedicina, University of Granada, 18016 Granda, Spain; 3Servicio de Microbiología, University Hospital Virgen de las Nieves, 18014 Granada, Spain; antonioj.sampedro.sspa@juntadeandalucia.es (A.S.); esther.gomez.vicente.sspa@juntadeandalucia.es (E.G.-V.); javierm.rodriguez.sspa@juntadeandalucia.es (J.R.-G.); jantonio.reguera.sspa@juntadeandalucia.es (J.A.R.); fernando.cobo.sspa@juntadeandalucia.es (F.C.); 4Departamento Bioquímica, Biología Molecular e Inmunología III, University of Granada, 18016 Granada, Spain; 5Instituto de Investigación Biosanitaria de Granada (ibs.GRANADA), 18012 Granada, Spain

**Keywords:** anti-S antibodies, HLA associations, mRNA-1273 vaccine

## Abstract

The vaccines designed against the SARS-CoV-2 coronavirus are based on the spike (S) protein. Processing of the S protein by antigen-presenting cells (APC) and its subsequent presentation to T cells is an essential part of the development of a humoral response. HLA-class II alleles are considered immune response genes because their codified molecules, expressed on the surface of APCs (macrophages, dendritic, and B cells) present antigenic peptides to T cell via their T cell receptor (TCR). The HLA-class II genes are highly polymorphic, regulating what specific peptides induce follicular helper T cells (TFH) and promote B lymphocyte differentiation into plasma or memory B cells. This work hypothesizes that the presence of certain HLA-class II alleles could be associated with the intensity of the humoral response (amount, length) to the SARS-CoV2 mRNA 1273 vaccine. We have studied the relationship between the HLA-class II typing of 87 health workers and the level of antibodies produced 30 days after vaccination. We show a possible association between the HLA-DRB1* 07:01 allele and the HLA-DRB1*07:01~DQA1*02:01~DQB1*02:02 haplotype to a higher production of antibodies 30 days after the administration of the second dose of mRNA-1273.

## 1. Introduction

Since December 2019, the rapid expansion of the SARS-CoV-2 coronavirus has resulted in a severe pandemic affecting the entire planet [1]. The countermeasures carried out, such as confinement, face masks, use of disinfectant gels, etc. have been of great help to combat the spread of the virus [2]. However, the development of vaccines against the virus is vitally important to avoid serious illness and death [3,4]. The mRNA-1273 vaccine employs messenger RNA (mRNA) technology and encodes a stabilized version of the SARS-CoV-2 full-length spike glycoprotein trimer [3]. The administration guideline requires two doses of 100 μg separated by 28 days [3].

Studies that monitor the cellular and humoral response of vaccinated people show different degrees of response depending on the vaccine [5,6]. Regarding antibody production, a good general response is observed at the beginning [7]. Subsequently, a decrease in the circulating level of anti-Spike (anti-S) antibodies is observed [8].

Differences in the antibody levels have been observed between individuals who received the same SARS-CoV-2 vaccine. Factors such as age, health, and immune system status intervene in the humoral response. In addition, immunogenetics can play an important role. The human leukocyte antigen class II (HLA-class II) molecules are part of the immunogenetic and perform antigen presentation during the generation of an immune response [9]. In the case of people vaccinated against SARS-CoV-2 (similar to other vaccines and infections), the spike (S) protein of SARS-CoV-2 generated artificially by the vaccine mRNA translation is taken up by dendritic cells and processed into peptides by the endosomal route. Those peptides are presented by HLA-class II molecules on dendritic cells to the T cell receptor (TCR) of T naive cells (Th0) to induce their differentiation to T follicular helper cells (Tfh) [10]. In the lymphoid follicle, B lymphocytes capture the S protein through their B-cell receptor (BCR), process it into peptides, and, as antigen-presenting cells (APC), present them in the context of HLA-class II molecules to the TCR of CD4 Tfh lymphocytes, forming the immunological synapse [10]. This results in the formation the germinal centers and the differentiation of B lymphocytes into SARS-CoV-2 Spike-specific plasma and memory B cells with isotype switch and affinity maturation via somatic hypermutations [11].

The HLA class II alleles that code for the HLA-DR, HLA-DQ, and HLA-DP molecules have a high degree of polymorphism. This polymorphism resides in the exons that code the cleft where the antigenic peptide is bound (β1 domain of HLA-DR and the α1 and β1 domains for HLA-DQ and HLA-DP molecules) [12]. Due to these polymorphisms, each allele can only present peptides that have binding motifs compatible to its specific cleft. For that reason, each allele presents different peptides derived from the same antigenic molecule. The presence of certain alleles will make the antigenic presentation more efficient, leading to a better stimulation of B cells that will mature into plasma cells. Likewise, the presence of HLA-class II alleles capable of presenting more peptides efficiently will form more specific clones of different epitopes of protein S, thus favoring a higher circulating antibody titer and probably a higher neutralizing and protective capacity. Therefore, the different HLA-class II alleles may explain the differences in antibody production observed between individuals.

In our work, we have studied the relationship between HLA class II polymorphism and humoral immunity generated by the SARS-CoV-2 mRNA-1273 vaccine in a cohort of 87 health workers from University Hospital Virgen de las Nieves. The comparison was made depending on the level of circulating IgG antibodies against protein S at 30 days after the second dose. 

## 2. Materials and Methods

**Population Studied**. The study was carried out with a population of 87 workers made up of 42 women and 45 men, belonging to the Hospital Universitario Virgen de las Nieves complex, who were vaccinated with Moderna’s mRNA-1273 vaccine. The mean general age was 48 years (23–65). 


**Control groups.**


-HLA control group: 637 healthy blood donors, representative of the Granada area, recollected between 2015 and 2021. The average age of the group is 45 years and 325 (51%) of the members are women. This group was used for compared the HLA-class II allelic frequencies in our region.

-Vaccinated control group: 601 workers made up of 398 women and 203 men, belonging to the University Hospital Virgen de las Nieves complex, who were vaccinated with Moderna’s mRNA-1273 vaccine. The mean general age was 48 years.

All patient samples were collected according to the local medical ethics regulation, after informed consent was obtained by the subjects, their legal representatives, or both, according to the Declaration of Helsinki. The studies involving human participants were reviewed and approved by Portal de Ética de la Investigación Biomédica. Junta de Andalucía (Cod. 0297-N-21). The patients/participants provided their written informed consent to participate in this study.

**Measurement of antibodies against SARS-CoV-2**. Participants underwent blood extraction, 30 days after inoculation of the second dose. A quantitative determination of immunoglobulin G (IgG) was performed against the S protein. The quantification of IgG was carried out by the chemiluminescent COVID-19 IgG Assay (Alinity, Abbott, IL, USA) following the manufacturer’s instructions. The results were expressed in BAU/mL (binding antibody units per milliliter). The cutoff for positivity was set at >7.5 BAU/mL. 

**Statistical Analysis**. Frequencies of individual HLA alleles and haplotypes were compared using the χ2-test. Variants with expected counts less than five were compared using Fisher’s exact test. Variants with an expected count of less than two were combined into a common class (binned) before computing the χ2-test. The software used was SPSS statistical software (Windows version 26, IBM, Armonk, NY, USA). Significance levels were corrected by Bonferroni correction for a multiplicity of testing by the number of comparisons. 

The Mann–Whitney U test was used to compare groups when the distribution was not normal (as checked by the Kolmogorov–Smirnov test).

A corrected ***p***-value of <0.05 was considered statistically significant.

**HLA typing by Next Generation Sequencing (NGS).** To amplify DNA target regions, we used the AllType FASTplex NGS 11 Loci Kit (One Lambda). For the HLA-DRB1/3/4/5 and HLA-DQB1 loci, the region between exon 2 and the 3’UTR region was amplified. The entire gene for the HLA-DQA1 locus was amplified. The technique was performed following the manufacturer’s recommendations. To load the chip, we used the Ion Chef (Thermo Fisher Scientific, Waltham, MA USA) and for sequencing the Ion GeneStudio S5 Plus System (Thermo Fisher Scientific). For data analysis, we used the TypeStream Visual NGS Analysis Software One Lamda (Thermo Fisher Scientific).

## 3. Results

**Quantification of anti-S protein antibody titers after SARS-CoV2 mRNA-1273 vaccination**. All the individuals studied presented specific anti-S IgG antibodies, 30 days after administration of the second dose of mRNA-1273 vaccine with a wide range: 65 to 10505 BAU/mL (Appendix A). The 87 cases were distributed in three groups. We based our classification on the distribution of anti-S antibodies levels in a cohort of 601 vaccinated individuals (Vaccinated control group). We use the mean value (2700 BAU/mL) +/- one standard deviation (SD) (1700 BAU/mL) to determine the groups (Appendix A). The 87 cases for the HLA class II typification were selected to achieve a similar number, sex frequency, and age in each group. The resulting humoral response groups were:-G1 (Low responders) (<1000 BAU/mL): 28 individuals (13 women and 15 men), with an average age of 46.8 years (26–65).-G2 (Middle responders) (1000–4400 BAU/mL): 29 individuals (14 women and 12 men), with an average age of 49.3 years (28–65).-G3 (High responders) (>4400 BAU/mL): 30 individuals (14 women and 16 men), with an average age of 48 years (23–65).

**HLA-class II allelic frequencies.** The HLA-class II alleles frequencies in the 87 cases were compared with an HLA control group of 637 blood donors, representative of the Granada area, without significant differences (Appendix A). HLA-DRB1 and HLA-DQB1 alleles frequencies in humoral response groups G1, G2, and G3 are represented in Table 1.

HLA-DRB1*07:01 and HLA-DQB1*02:02 alleles were more frequent in G3 vs G1 (high responders vs low responders). The *p*-value of HLA-DRB1*07:01 was potent and passed the Bonferroni correction (Pc = 0.0031), whereas HLA-DQB1*02:02 did not pass the Bonferroni correction (Table 2). HLA-DRB1*01:01 and HLA-DQB1*05:01 had a higher frequency in G1 but the *p*-value did not pass the Bonferroni correction. All the comparations with G2 for HLA-class II allelic frequencies were not significant.

Due to the previous result, we added the study of class II haplotypes. We detected 13 haplotypes with more than 2% of frequency (Appendix A). The HLA-DRB1*07:01~DQA1*02:01~DQB1*02:02 and HLA-DRB1*15:01~DQA1*01:02~DQB1*06:02 haplotypes were more frequent in G3, whereas HLA-DRB1*01:01~DQA1*01:01~DQB1*05:01 was more frequent in G1.

When the frequencies of HLA-class II haplotypes were compared between G1 and G3, HLA-DRB1*07:01~DQA1*02:01~DQB1*02:02 was statistically significant after correction with a higher frequency in G3 (Table 3). The frequencies of HLA-DRB1*01:01~DQA1*01:01~DQB1*05:01 and HLA-DRB1*15:01~DQA1*01:02~DQB1*06:02 did not show significance after Bonferroni correction (Table 3).

This result was in line with the allelic comparison and gives more value to HLA-DRB1*07:01.

Finally, we compared the mean anti-S antibody titers with to the presence or absence of HLA-DRB1*07:01, HLA-DRB1*01:01, and HLA-DRB1*07:01~DQA1*02:01~DQB1*02:02 in the volunteers studied (Figure 1). The presence of HLA-DRB1*07:01 was related to higher average levels of anti-S antibodies (*p* = 0.002), whereas the presence of HLA-DRB1*01:01 did not show significant differences. The comparison between cases with HLA-DRB1*07:01 versus cases with HLA-DRB1*01:01 showed a significant increase in anti-S anti-bodies in HLA-DRB1*07:01 carriers (*p*= 0.004). In addition, in cases with HLA-DRB1*07:01~DQA1*02:01~DQB1*02:02 we observed a higher anti-S antibody production versus subjects lacking this haplotype (*p* = 0.004) (Figure 1). 

We performed the comparison between HLA-DRB1*07:01 and HLA-DRB1*01:01 because of the results of the previous comparisons and because they are the alleles with the greatest frequency difference between groups. 

## 4. Discussion

The different production of antibodies against the S protein SARS-CoV-2 protein S in vaccinated people may be due to immunogenetic factors. Our results show that the HLA-DRB1*07:01 allele is significantly increased in the high responder’s group, whereas the HLA-DRB1*01:01 allele is close to significantly increased in the low responder’s group (Table 2). The HLA-DQB1*02:02 allele is close to being significant, being increased in high responders. This may be due to the haplotype that it forms together with the HLA-DRB1*07:01 allele. The haplotype study showed a significant increase for the HLA-DRB1 *07:01~DQA1*02:01~DQB1*02:02 haplotype in high responders. In addition, there is a significant difference in the mean production of anti-S antibodies in volunteers that are carriers of HLA-DRB1*07:01 and the haplotype HLA-DRB1*07:01~DQA1*02:01~DQB1*02:02, with a higher level of anti-S antibodies (Figure 1). HLA-DRB1*01:01 does not influence the anti-S antibody production, but the subjects with HLA-DRB1*01:01 have lower anti-S antibody titers than volunteers with HLA-DRB1*07:01 (Figure 1).

The different ability of distinct HLA-class II alleles to present peptides derived from the S protein is a possible explanation for the wide range of anti-S antibody production observed in our study. The HLA-DRB1*01:01 allele has a lower ability to strongly bind S protein-derived peptides compared to the HLA-DRB1*07:01 allele [13]. HLA-DRB1*01:01 presents only five peptides with high affinity, whereas HLA-DRB1*07:01 presents 16 [13]. In addition, the HLA-DRB1*01:01 allele lacks an associated HLA-DRB3, DRB4, or DRB5 molecule, having one less HLA-class II presenting molecule [14]. HLA-DRB1*07:01 presents an HLA-DRB4 molecule associated with the haplotype [14]. Hence, the HLA-DRB1*07:01~DQA1*02:01~DQB1*02:02, which is associated with a better response, may be able to activate a greater number of clones of distinct T cell clones resulting in a stronger humoral response, including higher production of antibodies against SARS-CoV-2 [9,15]. In contrast, the HLA-DRB1*01:01 allele and the HLA-DRB1*01:01~DQA1*01:01~DQB1*05:01 haplotype, have alleles with lower presentation capacity together with the absence of an HLA-DRB3, DRB4, or DRB5 molecule. This would induce a low stimulation of T cells, and therefore a lesser activation of B cells, leading to lower antibody production. In addition, HLA-DRB1*07:01~DQA1*02:01~DQB1*02:02 binds 30–34 peptides with high affinity, whereas HLA-DRB1*01:01~DQA1*01:01~DQB1*05:01 only binds to 16 peptides with high affinity. [13]. Moreover, HLA-DRB1*15:01 is the allele that presents more peptides with high affinity (26) [13] and it can be observed that its maximum frequency is found in high responders (Table 1). Moreover, HLA-DRB1*15:01~DQA1*01:02~DQB1*06:02, more frequent in high responders, binds to between 47 and 62 peptides with high affinity [13].

On the other hand, it is possible that the HLA-DRB1*01:01 allele is capable of generating a regulatory T cell response (Treg) that negatively regulates the immune response and stops further expansion and high antibody production [16]. However, this would be contrary to what is observed in patients with rheumatoid arthritis, where the presence of this allele poses a risk to the development of the disease with a dysregulation of the immune system [17]. From this, it can be thought that the HLA allele is as important as the peptide that is presented, observing differences in the induction and regulation of an immune response depending on it.

A factor that may explain why the HLA-DRB1*07:01 allele induces greater production of antibodies is that it presents an immunodominant peptide of protein S, which induces a greater differentiation of Tfh and high cooperation of B and T cells [18]. Related to this, it has been suggested that the HLA-DRB1*07:01 allele presents exogenous peptides that may favor the development of monoclonal TCR-Vβ13.1 + /CD4 + /NKa + /CD8−/+ dim T-LGL lymphocytosis by chronic stimulation [19].

The HLA-DRB1*07:01 allele has been linked as a risk factor for systemic lupus erythematosus (SLE) in the Malaysian population [20], whereas in the Chilean population, HLA-DRB1*07:01 had a protective role in anti-citrullinated protein antibodies-positive rheumatoid arthritis [21]. Moreover, HLA-DRB1*07:01 and the HLA-DRB1*07:01~DQA1*02:01~DQB1*02:02 haplotype are related to a higher risk of asparaginase hypersensitivity [22,23]. 

Finally, the presence of other possible polymorphisms in cytokines, receptors, etc., associated with the haplotype in the MHC region, may play an important role in the induction of the immune response and should be studied.

The results published by Ragone et al. showed that there is no relationship between the level of antibodies in the short or medium term with the number of peptides bound with high affinity in each individual vaccinated with BNT162b2 (BioNTech/Pfizer) [13]. They concluded that the level of antibodies is unrelated to HLA-class II molecules. In contrast, our results show a possible association of the HLA-DRB1*07:01 allele, and HLA-DRB1*07:01~DQA1*02:01~DQB1*02:02 haplotype, with a higher production of antibodies. The differences between both works are because they carry out a quantitative and theoretical study of the number of peptides that each allele presents with high affinity, whereas our study is based on traditional analysis of allele frequencies. However, it is possible that this different result is due to the vaccine studied but there are other reasonable explanations. For example, the excipients are different in the vaccines and the mRNA-1273 dose has a higher concentration of mRNA [24]. In addition, the mRNA-1273 vaccine induces a higher antibodies response than the BNT162b2 vaccine [24]. Finally, perhaps the mRNA-1273 vaccine induces a better stimulation of the immune system and higher antibody production that makes it possible to observe differences in the HLA-class II frequency.

In conclusion, our results show the first possible association between the circulating levels of anti-S antibodies induced by mRNA-1273 and HLA-class II alleles. However, complementary studies, a larger population will be necessary to confirm our results. In addition, it will be interesting to relate the neutralizing potential of the antibodies to the different HLA-class II alleles. Furthermore, will be interesting to compare the HLA-class II frequency between high and low responders in individuals vaccinated with mRNA-1273 versus individuals vaccinated with other vaccines. Finally, the study of the different peptides derived from protein S will be of great interest to clarify their immunological potential and the possible selection of immunodominant peptides.

Limitations of our work include the small population studied, not performing the neutralization study, and not comparing between vaccines. 

## Figures and Tables

**Figure 1 vaccines-10-00402-f001:**
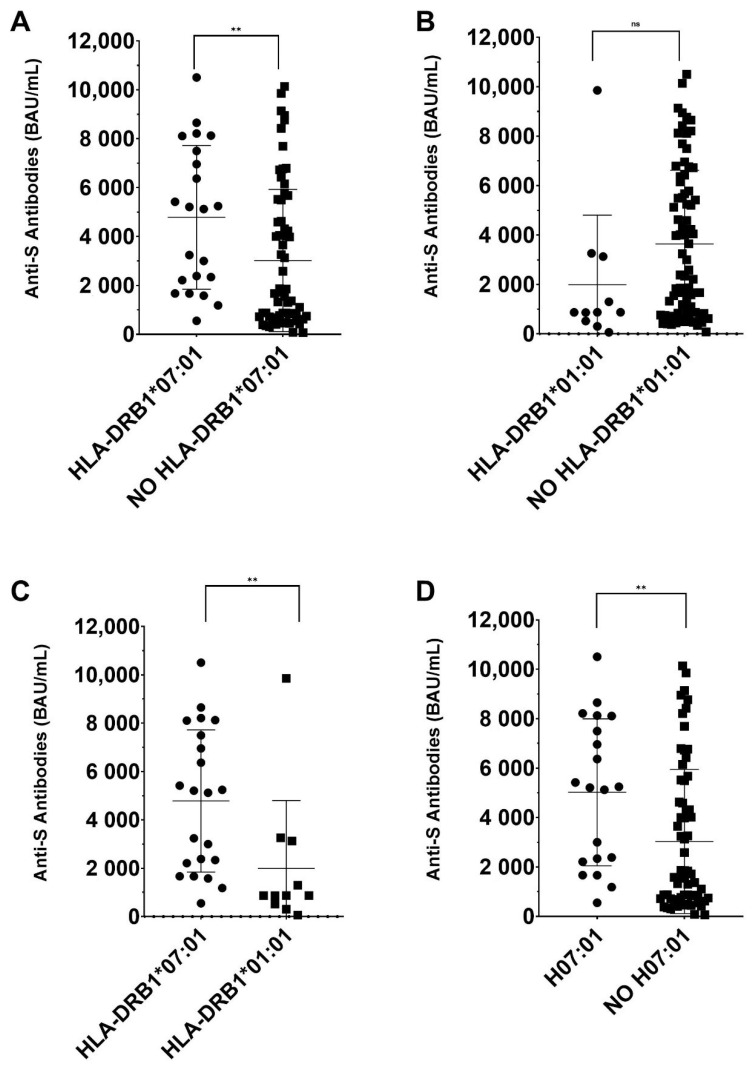
Anti-S antibodies mean comparations. Comparison of the mean production of anti-S antibodies depending on the presence or absence of certain alleles and haplotypes. (**A**) Cases with HLA-DRB1*07:01 versus cases without HLA-DRB1*07:01. (**B**) Cases with HLA-DRB1*01:01 versus cases without HLA-DRB1*01:01. (**C**) Cases with HLA-DRB1*07:01 versus cases with HLA-DRB1*01:01. (**D**) Cases with HLA-DRB1*07:01~DQA1*02:01~DQB1*02:02 haplotype (H07:01) versus cases without HLA-DRB1*07:01~DQA1*02:01~DQB1*02:02 haplotype (NO H07:01). ** = *p* < 0.01. ns= no significant.

**Table 1 vaccines-10-00402-t001:** Allele frequencies in the responder’s group.

Locus	Allele	Frequency		
		G1	G2	G3
HLA-DRB1	01:01	0.125	0.052	0.017
HLA-DRB1	01:02	0.107	0.034	0
HLA-DRB1	01:03	0.018	0	0.033
HLA-DRB1	03:01	0.071	0.172	0.067
HLA-DRB1	04:01	0	0.034	0.050
HLA-DRB1	04:02	0	0.034	0
HLA-DRB1	04:03	0.036	0	0.017
HLA-DRB1	04:04	0.054	0.052	0.017
HLA-DRB1	04:05	0.054	0	0.050
HLA-DRB1	04:06	0	0	0.017
HLA-DRB1	04:07	0.036	0	0
HLA-DRB1	07:01	0.018	0.172	0.250
HLA-DRB1	07:02	0	0	0.017
HLA-DRB1	08:01	0.018	0	0
HLA-DRB1	09:01	0.018	0.017	0
HLA-DRB1	10:01	0	0.017	0.017
HLA-DRB1	11:01	0.054	0.052	0.033
HLA-DRB1	11:02	0.018	0.034	0
HLA-DRB1	11:03	0.036	0	0
HLA-DRB1	11:04	0	0.052	0.050
HLA-DRB1	12:01	0.054	0	0
HLA-DRB1	13:01	0.071	0.069	0.100
HLA-DRB1	13:02	0.036	0.034	0.050
HLA-DRB1	13:03	0.018	0.017	0.017
HLA-DRB1	14:01	0.018	0	0
HLA-DRB1	14:54	0.018	0.034	0.017
HLA-DRB1	15:01	0.071	0.103	0.167
HLA-DRB1	16:01	0.054	0.017	0.017
HLA-DQB1	02:01	0.071	0.155	0.083
HLA-DQB1	02:02	0.036	0.121	0.200
HLA-DQB1	02:05	0	0.017	0
HLA-DQB1	02:10	0	0.017	0
HLA-DQB1	03:01	0.196	0.155	0.117
HLA-DQB1	03:02	0.179	0.103	0.117
HLA-DQB1	03:03	0	0.052	0.050
HLA-DQB1	03:19	0.018	0.034	0
HLA-DQB1	04:02	0.018	0	0.017
HLA-DQB1	05:01	0.232	0.086	0.067
HLA-DQB1	05:02	0.054	0.017	0.017
HLA-DQB1	05:03	0.036	0.034	0.017
HLA-DQB1	06:01	0	0.034	0
HLA-DQB1	06:02	0.071	0.069	0.150
HLA-DQB1	06:03	0.054	0.086	0.117
HLA-DQB1	06:04	0.018	0.017	0.017
HLA-DQB1	06:09	0.018	0	0.033

HLA-DRB1*01:01 and HLA-DQB1* 05:01 were more frequent in G1; HLA-DRB1*03:01, HLA-DRB1*07:01, HLA-DQB1* 02:01 and HLA-DQB1*03:01 in G2 and HLA-DRB1*07:01 and HLA-DQB1*02:02 in G3.

**Table 2 vaccines-10-00402-t002:** Comparison of allele frequencies.

Allele	Frequency		P	Pc
	Low Responders	High Responders		
HLA-DRB1*01:01	**0.125**	0.017	0.028	n.s
HLA-DQB1*05:01	**0.232**	0.067	0.016	n.s
HLA-DRB1*07:01	0.018	**0.250**	**2.3 × 10^−4^**	**3.1 × 10^−3^**
HLA-DQB1*02:02	0.036	**0.200**	0.008	n.s

The highest frequency for each allele is marked in bold. OR: odds ratio. Pc: P corrected by Bonferroni. n.s: not significant.

**Table 3 vaccines-10-00402-t003:** Comparison of haplotype frequencies.

Haplotype	Frequency in Low Responders	Frequency in High Responders	P	Pc
HLA-DRB1*01:01~DQA1*01:01~DQB1*05:01	**0.107**	0.017	n.s	n.s
HLA-DRB1*07:01~DQA1*02:01~DQB1*02:02	0.018	**0.200**	2.1× 10^−3^	**0.028**
HLA-DRB1*15:01~DQA1*01:02~DQB1*06:02	0.054	**0.133**	n.s	n.s

The highest frequency for each allele is marked in bold. OR: odds ratio. Pc: P corrected by Bonferroni. n.s: not significant.

## Data Availability

Not applicable.

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
