# Peer review of "HLA Class II Polymorphism and Humoral Immunity Induced by the SARS-CoV-2 mRNA-1273 Vaccine"

_vaccines, 2022, doi:10.3390/vaccines10030402_

Round 1
Reviewer 1 Report
The manuscript of Gutiérrez-Bautista et al. presents a minor and insignificant piece of data with respect of IgG antibody formation in response to vaccination with Moderna SARS-CoV-2 vaccine at a single time-point (30 days after the 2nd dose) and using a very limited number of test subjects (a total of 87). The only result that barely passes statistical significance is supposedly higher frequency of HLA-DRB1*07:01~DQA1*02:01~DQB1*02:02 allele in high responders, but the value of this observation is highly dubious since the whole premise of initial separation of test subjects into low, medium and high responders is not clear and the distribution of subjects within all of these groups (e.g., mean titer +/- SD) is nowhere to be seen. It is more than likely that even a slight shift in arbitrary group borders created by the authors would result in a different outcome (it is not surprising then that instead of giving of raw numbers of subjects carrying for each allele they opt for 'frequency', the calculation of which, in turn, is never properly described). In essence, authors have nothing to report even if they try to present otherwise. In addition, manuscript is poorly written and poorly organized with both Introduction and Discussion having very little relevance to its subject, the latter one been completely overblown related to experimental observations made. Moreover, there is a number of improperly cited references, e.g. Ref. 7 is cited as describing the immune response after vaccination, while it describes the immune response after Covid-19 infection and Ref. 9 is cited as if it describes the immune response to S protein of SARS-CoV-2, while it discusses general mechanisms of antigen presentation and was published in 2014. It also doesn't help that the manuscript is poorly written, e.g. its title is already incorrect (should have been 'in INDIVIDUALS vaccinated', etc.) and contains many hardly legible sentences, e.g. 'Due to these polymorphisms, each allele can only bind to peptides that have binding motifs capable of binding to the specific cleft of each allele' followed by another sentence that is difficult to understand: 'Two different alleles have different peptide groups even if they are the same antigenic molecule' (lines 57-60). Collectively, this is a limited, insignificant, poorly-described and poorly-presented piece of data that hardly adds anything currently known with respect to humoral response induced by vaccination to SARS-CoV-2 S protein.
Author Response
Dear Reviewer,
Thank you for your comment.
We had made some changes in the manuscript that improve our work.
The manuscript of Gutiérrez-Bautista et al. presents a minor and insignificant piece of data with respect of IgG antibody formation in response to vaccination with Moderna SARS-CoV-2 vaccine at a single time-point (30 days after the 2nd dose) and using a very limited number of test subjects (a total of 87). The only result that barely passes statistical significance is supposedly higher frequency of HLA-DRB1*07:01~DQA1*02:01~DQB1*02:02 allele in high responders, but the value of this observation is highly dubious since the whole premise of initial separation of test subjects into low, medium and high responders is not clear and the distribution of subjects within all of these groups (e.g., mean titer +/- SD) is nowhere to be seen.
We add an explanation of our group distribution in Results.
It is more than likely that even a slight shift in arbitrary group borders created by the authors would result in a different outcome (it is not surprising then that instead of giving of raw numbers of subjects carrying for each allele they opt for 'frequency', the calculation of which, in turn, is never properly described).
In essence, authors have nothing to report even if they try to present otherwise. In addition, manuscript is poorly written and poorly organized with both Introduction and Discussion having very little relevance to its subject, the latter one been completely overblown related to experimental observations made.
Moreover, there is a number of improperly cited references, e.g. Ref. 7 is cited as describing the immune response after vaccination, while it describes the immune response after Covid-19 infection and Ref. 9 is cited as if it describes the immune response to S protein of SARS-CoV-2, while it discusses general mechanisms of antigen presentation and was published in 2014.
The references have been checked.
It also doesn't help that the manuscript is poorly written, e.g. its title is already incorrect (should have been 'in INDIVIDUALS vaccinated', etc.) and contains many hardly legible sentences, e.g. 'Due to these polymorphisms, each allele can only bind to peptides that have binding motifs capable of binding to the specific cleft of each allele' followed by another sentence that is difficult to understand: 'Two different alleles have different peptide groups even if they are the same antigenic molecule' (lines 57-60).
We edited the manuscript to do a better redaction.
English revision has been done.
Collectively, this is a limited, insignificant, poorly-described and poorly-presented piece of data that hardly adds anything currently known with respect to humoral response induced by vaccination to SARS-CoV-2 S protein.
We hope that the work has been improved and can be considered for publication.

Reviewer 2 Report
In the manuscript "HLA-DRB1*07:01~DQA1*02:01~DQB1*02:02 haplotype is related to a higher production of antibodies in vaccinated with Moderna" by Juan Francisco Gutiérrez-Bautista and colleagues, the authors studied the relationship between the HLA-class II typing of 87 health workers and the level of antibodies produced 30 days after Moderna´s COVID-19 vaccine.
I have some major concerns about this manuscript:
1) Lines 220-200: I don´t agree this study shows a clear association of the HLA-DRB1*07:01 allele, and HLA-DRB1*07:01~DQA1*02:01~DQB1*02:02 haplotype, with a higher production of antibodies, because all alleles must be included in the statistical analysis in order to obtain significance. This conclusion is incorrect and must be removed.
2) The allele frequency is not the only parameter of relevance for antigen presentation. The binding affinity of peptide-MHC complex is a very important parameter that must be considered, before any conclusions are drawn.
3) Please make sure the supplementary data is available for the reviewers. A decision can only be taken after all the evidence has been taken into consideration.
Given these 3 objections, the manuscript should not be accepted in the current format. The manuscript should undergo a major revision before being considered for acceptance.
Minor concerns:
4) Title: Please refer to the correct reference of Moderna’s COVID-19 vaccine.
5) Line 25: The sentence “30 days after the administration of the second dose of Moderna” is imprecise. Please refer to the correct reference of Moderna’s COVID-19 vaccine.
6) Lines 35-37: The sentence “In general, a good response is observed which in the case of the antibody level decreases to a plateau phase where it is maintained [7].”, is not very clear. Please rephrase it.
7) Lines 80 and 82: Please replace “patients” with volunteers.
8) Line 151: What do the authors mean with “potency”? Please elaborate.
9) The information shown in lines 129-130, was already shown in the Materials and Methods section, lines 91-96. This is a redundancy and must be corrected.
10) Line 134: When introducing the concept of “resolution”, please describe it clearly and elaborate on its relevance.
11) Line 142: Please remove the sentence “HLA-DR1 is really close to significant (Pc=0.063)”. HLA-DR1 is not significant.
12) In the tables 1 and 2 please don´t be redundant by repeating the word “frequency” 4 times. Please show it only once.
13) Line 219: Please include the name of the vaccine and not “Pfizer” alone.
14) The results section is very simplistic, not explaining the rationale applied to analyse the data. The authors must explain clearly the rationale used during the presentation of results.
15) Is HLA-DRB1*07:01 associated with susceptibility to any disease known? Please elaborate on this point and include it in the Discussion section.
16) Please include a new figure, showing a correlation between the IgG titres and the allele frequency.
17) The manuscript must be reviewed by a native English speaker.
Author Response
Dear Reviewer,
Thank you for your valuable comments.
- Lines 220-200: I don´t agree this study shows a clear association of the HLA-DRB1*07:01 allele, and HLA-DRB1*07:01~DQA1*02:01~DQB1*02:02 haplotype, with a higher production of antibodies, because all alleles must be included in the statistical analysis in order to obtain significance. This conclusion is incorrect and must be removed.
We removed this incorrect conclusion. The title has been change because it is more appropriate for our work.
- The allele frequency is not the only parameter of relevance for antigen presentation. The binding affinity of peptide-MHC complex is a very important parameter that must be considered, before any conclusions are drawn.
We add this important parameter in the discussion section. Thank you for your recommendation.
- Please make sure the supplementary data is available for the reviewers. A decision can only be taken after all the evidence has been taken into consideration.
We are sorry. Maybe a mistake occurred. We hope that the new Supplementary Material will be available.
- Title: Please refer to the correct reference of Moderna’s COVID-19 vaccine.
We changed the title and added the correct reference of Moderna’s COVID-19 vaccine.
- Line 25: The sentence “30 days after the administration of the second dose of Moderna” is imprecise. Please refer to the correct reference of Moderna’s COVID-19 vaccine.
We add the correct reference of Moderna´s COVID-19 vaccine (Line 19 and 23 of the new version).
- Lines 35-37: The sentence “In general, a good response is observed which in the case of the antibody level decreases to a plateau phase where it is maintained [7].”, is not very clear. Please rephrase it.
We have rewritten and expressed that sentence more clearly (Lines 36 to 38 of the new version).
- Lines 80 and 82: Please replace “patients” with volunteers.
We did this change.
- Line 151: What do the authors mean with “potency”? Please elaborate.
We eliminated this incorrect concept.
- The information shown in lines 129-130, was already shown in the Materials and Methods section, lines 91-96. This is a redundancy and must be corrected.
This part has been corrected.
- Line 134: When introducing the concept of “resolution”, please describe it clearly and elaborate on its relevance.
We eliminated this concept because we used only one resolution finally.
- Line 142: Please remove the sentence “HLA-DR1 is really close to significant (Pc=0.063)”. HLA-DR1 is not significant.
This sentence has been removed.
- In the tables 1 and 2 please don´t be redundant by repeating the word “frequency” 4 times. Please show it only once.
We have fixed this bug
13) Line 219: Please include the name of the vaccine and not “Pfizer” alone.
The correct name of the Pfizer COVID-19 vaccine has been added (Line 254 of the new versión)
- The results section is very simplistic, not explaining the rationale applied to analyse the data. The authors must explain clearly the rationale used during the presentation of results.
We have changed the results section. We explain the result more clearly and why we do this analysis.
- Is HLA-DRB1*07:01 associated with susceptibility to any disease known? Please elaborate on this point and include it in the Discussion section.
We added in the discussion section the association of HLA-DRB1*07:01 with diseases (Lines 243 to 248 of the new version).
- Please include a new figure, showing a correlation between the IgG titres and the allele frequency.
We added a figure with relation to the presence of certain alleles with the mean of anti-S antibodies. We add the comparison between the mean of antibodies depending on the presence or absence of determinant alleles or haplotypes.
- The manuscript must be reviewed by a native English speaker.
English revision has been done.
We edited the manuscript to do a better redaction for the recommendation of the first reviewer.
Thank you for your interesting comments.
We hope that the work has improved and can be considered for publication.
Round 2
Reviewer 1 Report
This is a much-improved version of the original manuscript. Although not a major contribution to the field, it can be published in the current form, although another round of English language revision is required (e.g., 'Differences in the antibodies levels' in line 39 instead of 'Differences in the antibody levels', etc.)
Author Response
This is a much-improved version of the original manuscript. Although not a major contribution to the field, it can be published in the current form, although another round of English language revision is required (e.g., 'Differences in the antibodies levels' in line 39 instead of 'Differences in the antibody levels', etc.)
Dear Editor,
Thank you for your valuation.
We did another round of English language revision.
We hope that the manuscript is ready for publication.
King Regards,
Juan Francisco Gutiérrez-Bautista

Reviewer 2 Report
Line 115 - In the first paragraph of the Results section, there is a description of "anti-S antibodies levels in a cohort of 601 individuals". However, in the Materials and Methods section the authors describe using data from 87 volunteers only. What is this cohort of 601 individuals?
Could to authors please be very clear in the number of human subjects used?
It is extremely important to refer to all human subjects used in any study.
Please make sure to refer to the proper Institutional Ethical Approval and Informed Consent from all 601 volunteers used in this study.
Line 127 - The study refers to data obtained from "637 blood donors". Again, please be very clear with the proper Institutional Ethical Approval and Informed Consent from all volunteers used in this study.
Lines 118-119: Please remove "IN EACH GROUP" in capitals.
Line 143 - please remove "in high resolution".
Author Response
Reviewer 2
Dear Editor,
Thank you for your quick response.
Line 115 - In the first paragraph of the Results section, there is a description of "anti-S antibodies levels in a cohort of 601 individuals". However, in the Materials and Methods section the authors describe using data from 87 volunteers only. What is this cohort of 601 individuals?
We added the description of this cohort in Materials and Methods (Line 85 of the new version)
Could to authors please be very clear in the number of human subjects used?
We have been changed the description of the subject used. We did a mistake in the before revision and we did not add the controls cohort in Materials and Methods. We are sorry for the mistake.
It is extremely important to refer to all human subjects used in any study.
In the Materials and Methods section, there is a new description of all cohorts used for this study. (Line 80 of the new version)
Please make sure to refer to the proper Institutional Ethical Approval and Informed Consent from all 601 volunteers used in this study.
Have been added the Institutional Ethical Approval and the Informed Consent in the Material and Methods section (Line 88 of the new version).
Line 127 - The study refers to data obtained from "637 blood donors". Again, please be very clear with the proper Institutional Ethical Approval and Informed Consent from all volunteers used in this study.
In the Materials and Methods section have been added the description of all cohorts, the Institutional Ethical Approval, and Informed Consent (Lines 80-94 of the new version).
Lines 118-119: Please remove "IN EACH GROUP" in capitals.
This mistake has been corrected. (Line 131 of the new version)
Line 143 - please remove "in high resolution".
This mistake has been corrected (Lines 157 of the new version).
We did another round of English language revision.
We hope that the manuscript is ready for publication.
King Regards,
Juan Francisco Gutiérrez-Bautista
Round 3
Reviewer 2 Report
I checked the last version, containing all edits by the authors. I am happy to reconsider my review and accept the manuscript in the current form.